# Digitally delivered, systemically challenged: A qualitative study of health system readiness for digital care

Laurel O'Connor[1]*, Leah Dunkel[1], Andrew C. Weitz[2], Allan Walkey[1], Peter K. Lindenauer[3], Apurv Soni[1]

**1** Program in Digital Medicine, Department of Medicine, University of Massachusetts Chan Medical School, 55 Lake Avenue North, Worcester, Massachusetts, United States of America, **2** National Institute of Biomedical Imaging and Bioengineering (NIBIB), National Institutes of Health (NIH), Rockville Pike, Bethesda, Maryland, United States of America, **3** Department of Healthcare Delivery and Population Sciences and Department of Medicine, University of Massachusetts Chan Medical School–Baystate, Springfield, Massachusetts, United States of America

☙ Andrew Weitz contributed to this work when he was an NIH employee; however, he is no longer affiliated with the NIH.

\* laurel.oconnor@umassmed.edu

## Abstract

Digital health technologies (DHTs) expand healthcare access, improve care coordination, and reduce costs. However, integrating these tools into care faces complex barriers. Understanding the perspectives of health system leaders is essential for developing sustainable DHTs. The objective of this project is to explore the experiences and priorities of health system stakeholders regarding the implementation of DHTs. The study team conducted semi-structured interviews with 12 stakeholders from diverse U.S. health systems, including clinical, operational, and executive leadership. Interviewees were selected using purposeful and snowball sampling. Interviews were transcribed and analyzed thematically using the Consolidated Framework for Implementation Research (CFIR). A constant comparative coding process was used to identify and organize key themes. Participants viewed DHTs as a way to enhance healthcare access and efficiency and improve public health operations, especially in rural or underserved settings. However, several major adoption challenges emerged: (1) integrating DHTs into existing workflows and electronic health records is operationally burdensome; (2) digital care can introduce risks to quality, continuity, and equity; and (3) external factors (reimbursement policy, regulatory constraints, infrastructure investment) are critical to long-term adoption. Digital health is seen as essential to the future of healthcare delivery, but meaningful integration requires alignment across clinical, operational, and policy domains. Coordinated investment, regulatory reform, and robust data infrastructure are needed to ensure DHTs are scalable and sustainable.

**Data availability statement:** All relevant data are within the manuscript and its Supporting Information files.

**Funding:** This work was supported by the National Institutes of Health, National Institute of Biomedical Imaging and Bioengineering (Contract No. 75N92022D00010 to AS) and National Institutes of Health, National Heart, Lung, and Blood Institute (1K23HL174454-01A1 to LO) The funders had no role in study design, data collection and analysis, decision to publish, or preparation of the manuscript.

**Competing interests:** The authors have declared that no competing interests exist.

## Author summary

This study explored how senior leaders from diverse U.S. health systems view the opportunities and challenges of digital health technologies (DHTs), such as telehealth and remote patient monitoring. Through in-depth interviews, we found that health system leaders see DHTs as promising tools to improve access to care, efficiency, and public health, especially in rural and underserved areas. However, they identified several barriers to successful adoption, including difficulties integrating DHTs with existing workflows and electronic health records, concerns about data privacy and cybersecurity, and the potential for fragmented or inequitable care. Leaders emphasized that factors such as reimbursement policies, regulatory requirements, and investment in digital infrastructure play a crucial role in the long-term sustainability of DHTs. Without coordinated actions across healthcare, policy, and technology sectors, digital health innovation risks exacerbating disparities and failing to be sustainable. The findings highlight that bridging the digital divide and establishing supportive national policies are key to making digital health a lasting part of routine healthcare, rather than isolated pilot programs. This research provides actionable insights for policymakers, healthcare organizations, and technology developers aiming to deliver equitable and effective digital health solutions.

## Introduction

Digital health technologies (DHTs) are increasingly common in clinical care, providing unprecedented access: patients can undergo pulmonary rehabilitation, get tested for hepatitis C, monitor depression symptoms, receive diabetes coaching, and even obtain COVID-19 evaluation and treatment – all without leaving their home [1–6]. Emerging technologies offer promising solutions to longstanding challenges in access, efficiency, and patient engagement [7]. Smartphones, wearable devices, remote monitoring tools, and telehealth platforms enable the delivery of care beyond traditional brick-and-mortar clinical settings [8]. Accelerated by the COVID-19 pandemic, these technologies are now integrated into care models, allowing providers to extend services into patients' homes and foster continuous, data-driven care [9,10]. Importantly, DHTs hold particular promise for addressing disparities in access by reaching rural, homebound, and underserved populations who may otherwise face significant barriers to in-person care [11–13].

Integrating DHTs into traditional care presents numerous challenges. Key issues include 1) a lack of standard interoperability between DHTs and electronic health record (EHR) systems, 2) the required changes to established care processes, necessitating provider training, workflow redesign, and institutional buy-in, 3) lack of standardized strategies governing implementation and maintenance of DHTs to determine the security of the data, quality of solutions, and their comparability with each-other and/or analog comparator [14], 4) differences in "techquity" and digital

literacy among patients that are closely linked to social determinants of health, including income, education, structural racism, and access to broadband and devices [13,15 and 5] reimbursement uncertainty [16]. Addressing these challenges requires coordinated efforts across technology developers, healthcare institutions, and policymakers to ensure that digital innovations are seamlessly and sustainably embedded into routine practice [16].

Implementation science best practices emphasize identifying implementation barriers and facilitators from key stakeholder perspectives [17]. System-level stakeholders (e.g., health systems, payors, policymakers) often serve as gatekeepers to real-world adoption [17]. Thus, their priorities and decision-making frameworks must be considered in the design and dissemination of DHTs [17]. The objective of this study is to explore health system stakeholders' perspectives on integrating DHTs into clinical practice.

## Methods

### Defining digital health technologies

For this project, DHTs were defined as a broad set of technologies that support the delivery, coordination, and monitoring of healthcare services outside traditional clinical settings [18,19]. These include tools such as telehealth platforms, remote patient monitoring devices, mobile health applications, wearable sensors, digital diagnostics (e.g., at-home testing models), and integrated data platforms [18,19]. In this context, "integration" of DHTs refers not only to technical EHR interoperability but also to the broader alignment of workflows, teams, and system processes needed for seamless clinical use.

### Study design and setting

This qualitative study aimed to identify determinants for adopting DHTs by leveraging the implementation science methodology of CENTERing multi-level partner voices in Implementation Theory (CENTER-IT), which is grounded in the more widely known Consolidated Framework for Implementation Research (CFIR) [20,21].The CENTER-IT approach draws on CFIR's validated domains for understanding what facilitates or hinders intervention implementation but introduces a structured strategy that actively involves stakeholders at multiple levels, including clinicians, operational leaders, and system administrators, in identifying barriers and shaping practical solutions. In this way, CENTER-IT transforms CFIR's broad determinants into a structured, collaborative process for adapting interventions so they better fit real-world workflows, policies, and system constraints. The goal of this approach is to generate actionable insights that support successful program development, scalability, and long-term adoption [20,21]. The CENTER-IT approach consists of four structured phases: (1) conducting interviews with both recipients and deliverers of the intervention to identify key implementation barriers; (2) selecting expert partners operating within the inner and outer contexts of the intervention to address the identified barriers; (3) engaging these expert stakeholders in facilitated discussions to explore barriers and co-develop potential solutions; and (4) adapting the intervention based on synthesized input across stakeholder levels [20]. Notably, this methodology is specifically intended for the implementation and adoption of existing evidence-based interventions, with the investigative focus centered on overcoming contextual challenges rather than modifying the intervention itself [20].

The study staff were based at an urban academic tertiary care medical center in Massachusetts that serves urban, suburban, and rural communities. This study complied with the consolidated criteria for best practices in reporting qualitative research [22,23] and was approved by the Institutional Review Board of the affiliate medical school (IRB 00004879), as well as Western Copernicus Group (IRB # 1355598).

### Selection of participants

Participants were recruited from geographically diverse U.S. regions using hybrid purposeful and snowball sampling. We used a purposive strategy to identify stakeholders across clinical, operational, and administrative roles relevant to digital care delivery, drawing from both academic and community-based settings to capture variation in digital infrastructure, resource availability,

and organizational priorities. The team initially identified key stakeholder roles (e.g., chief quality, executive, informatics, medical officers) based on literature and expert input, then expanded this list during recruitment [8,17,20]. The team sought diversity in professional background and system context to ensure representation across components of the health system influencing digital care integration. Prior experience with digital health was not required [24,25]. Snowball sampling facilitated a robust exploration of implementation challenges and ensured thematic saturation [26]. Recruitment was done via direct outreach and email, with a nominal gift card offered for participation, and ceased once thematic saturation was achieved.

## Data collection

The interview guide was structured by CFIR domains, with sections related to intervention characteristics, individuals involved, inner setting, and outer setting. Its development was informed by the study team's prior research, as well as relevant findings from current literature [21]. CFIR was selected due to its comprehensive scope, adaptability to diverse interventions, and its central role within the CENTER-IT methodology [21]. The guide was piloted with two "test" individuals, after which revisions were made to optimize clarity, relevance, and flow before use with enrolled participants.

Interviews commenced with a brief description of a specific digital health intervention, the Home Test to Treat program [27]. The interviewer described the intervention, which entailed home-based testing, telehealth services, and antiviral treatment for influenza and COVID-19. The interviewer emphasized that patients were able to access testing, medical evaluation, and treatment for these viral infections completely independently from brick-and-mortar clinical settings, but that there was no information exchange between the program and the patients' medical records. Participants were asked to discuss their impressions of the opportunities and risks of implementing such a program in their own system. They were then asked a series of open-ended questions about their perspectives more generally on the implementation and integration of DHTs. The Home Test to Treat model was included to provide participants with a shared point of reference; however, interviewees were encouraged to extrapolate beyond this intervention to consider broader DHTs, including remote monitoring, asynchronous communication tools, and system-level digital workflows.

Questions were organized into three categories: (1) health system integration for digital programs, (2) any existing strategic approach to DHTs, and (3) DHT implementation processes. Within each domain, CFIR constructs related to intervention characteristics, inner and outer settings, and individual-level factors were used to guide question development. The guide was refined iteratively throughout the study period in response to interviewer feedback and emerging themes; the finalized version is available in Supplemental File S1 Appendix.

Participants received a study fact sheet during recruitment and again immediately before their interview. The fact sheet outlined the interview's purpose, structure, and potential risks and benefits of participation. Interview questions were not provided in advance. Semi-structured interviews were conducted privately via video teleconferencing between October 2024 and April 2025. Each session lasted approximately 60 minutes and was led by one or two physician investigators with experience in digital health implementation, both of whom had prior training and experience in qualitative methods and structured interviewing. Interviews were audio recorded and transcribed using Otter.ai artificial intelligence software (Mountain View, California, United States). Transcripts were reviewed by a member of the research team to verify accuracy and ensure the removal of all identifying information.

## Analysis

Transcripts were imported into Atlas.ti (v23.2, Berlin, Germany) for qualitative coding. The research team developed an initial top-level codebook based on prior literature and CFIR domains. Two researchers independently coded five randomly selected transcripts using the initial codebook. In parallel, the team used an iterative constant-comparative approach to identify recurring patterns and inductively capture new themes, which were incorporated into the codebook. After the first five transcripts, Krippendorff's alpha was 0.90, indicating high inter-coder reliability [28].

Three team members (two physicians and one research scientist) coded all transcripts. Because the physician inter-viewers had prior experience with digital health delivery and implementation, the team engaged in reflexive practices throughout analysis, including explicitly discussing assumptions, documenting analytic decisions, and incorporating the perspectives of the non-clinical research scientist to balance potential interpretive bias. Each transcript was coded by two members independently, and then the team met to review and reach consensus on codes. After each meeting, transcripts were re-coded with the refined codebook. This process continued until no new themes emerged, indicating thematic satu-ration [29].

## Results

In total, 13 potential participants were approached, and 12 participated (92.3% response rate). Table 1 summarizes the professional titles of each participant. Several prevalent themes emerged during the interviews, which were grouped and organized hierarchically. Fig 1 provides a summary of elicited themes, organized by CFIR domain.

### Theme 1: Integrating DHTs into Existing Clinical Workflows Presents Significant Informatics and Cybersecurity Challenges

Participants emphasized the operational complexity of integrating DHTs into existing workflows (Table 2). Clinical teams are expected to take on new tasks, such as retrieving and interpreting data from external digital sources, and incorporate that information into their decision-making, often without additional staff, space, or reimbursement. This burden is com-pounded by technical issues: many DHTs are not interoperable with standard clinical software, forcing staff to use inef-ficient workarounds. The lack of seamless integration was seen as a major barrier because of the extra cognitive load it creates.

A major contributor to the logistical obstacles to integrating third-party DHTs is the lack of meaningful interoperability across EHR platforms. This was largely attributed to vendors' business models. While federal efforts to promote interoper-ability, such as Meaningful Use incentives, were acknowledged, participants expressed frustration that these changes had not translated into functional, real-world data exchange. Others called for a national infrastructure to govern and standard-ize digital information sharing, comparing the need for oversight to that of "highways" or "air traffic control." Participants' descriptions of fragmented systems and poor interoperability reflect barriers related to the CFIR construct of compatibility and complexity, emphasizing the lack of alignment between DHTs and existing workflows. Many perceived these misalign-ments as adding work rather than streamlining care, thus also demonstrating high complexity from the user perspective.

**Table 1. Participant Roles.**

| Participant Number | Role | Region |
|---|---|---|
| 1 | Chief Clinical Informatics Officer | Urban northeast |
| 2 | Chief Executive Officer | Urban northeast |
| 3 | Executive Vice President and Chief Financial Officer | Urban northeast |
| 4 | Division Chief, General Internal Medicine | Urban northeast |
| 5 | Chief Quality Officer | Rural northeast |
| 6 | Medical Group Associate Medical Director | Rural northeast |
| 7 | Chief Medical Officer | Urban and rural midwest |
| 8 | Health System President | Urban northeast |
| 9 | Chief Medical Information Officer | Urban northeast |
| 10 | Chief Medical Officer | Rural south |
| 11 | Chief Financial Officer | Rural south |
| 12 | Chief Nursing Officer | Rural south |

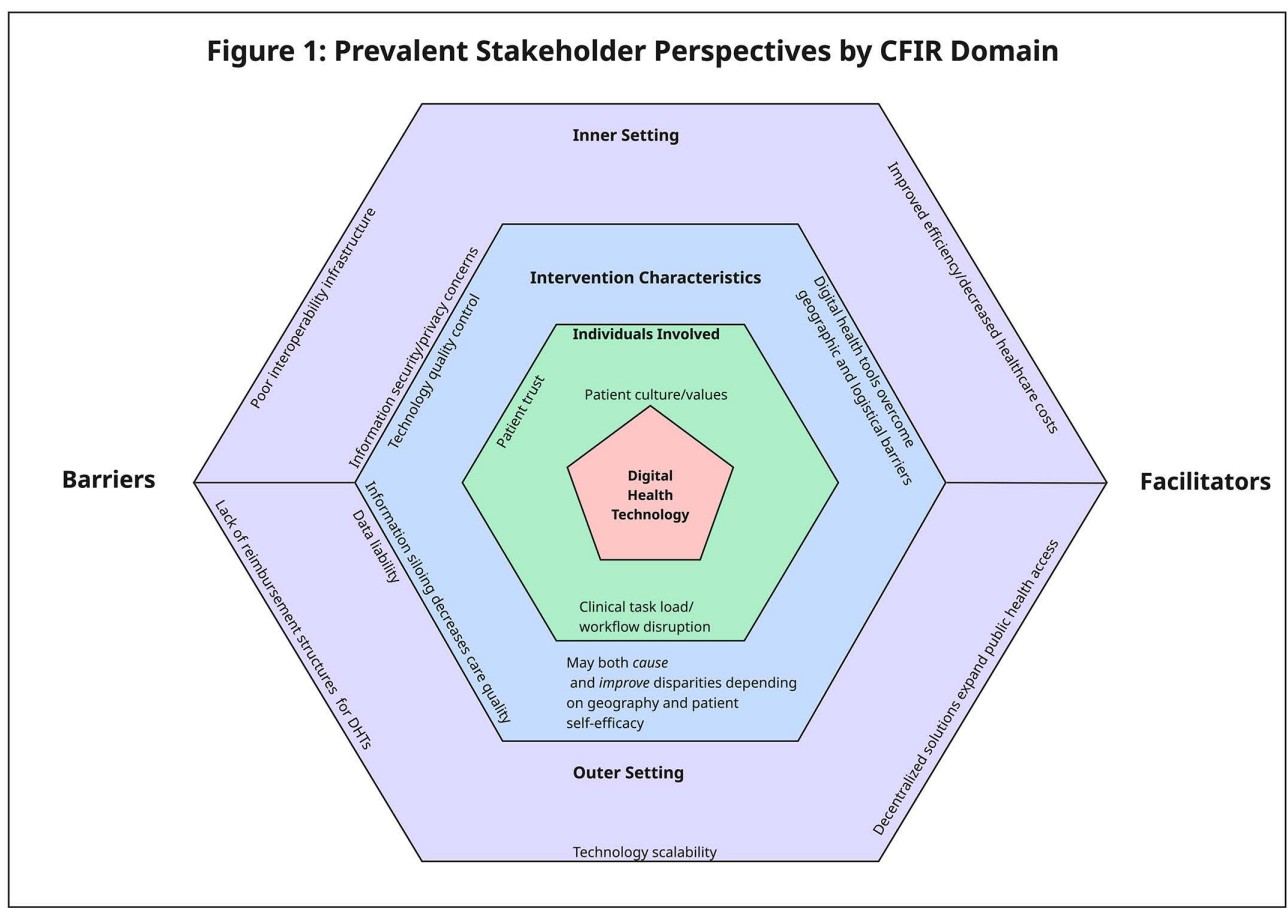

**Fig 1. Prevalent Stakeholder Perspectives by CFIR Domain.** The figure highlights domain-specific barriers and facilitators influencing adoption of the digital health technology, including patient- and clinician-level factors, workflow considerations, organizational context, and external influences.

Finally, participants universally expressed concerns about legal risks and data oversight. There was deep concern over cybersecurity vulnerabilities, particularly when third-party vendors were involved in data exchange. Many participants emphasized that while internal cybersecurity protocols were strong, external partners posed a potential risk that health systems would assume when allowing information delivery. There was also concern about risk and liability relating to the management of dynamic data streams of unclear clinical meaning with regard to monitoring and interpretation. A participant noted

"One problem…with the provider buy-in- let's say the patient has a blood sugar of 600. And maybe that smart meter… is attached to a cell phone tower, but maybe that meter reading didn't come to our EMR. It didn't alert us. And so, there's a possibility of that being overlooked. Maybe the patient has symptoms and goes to the hospital, but we never receive that alert. So, there's…concern for liability, yes, and proactive monitoring…Maybe we're not sure whether all the bugs are out of it. So right now, I think we see it more as a risk than a benefit." (Participant 12, Chief Nursing Officer)

These perspectives align with CFIR inner setting factors, including networks and communications, compatibility, culture, and readiness for implementation, where inconsistent data pathways and unclear ownership created uncertainty about how, and by whom, digital data should be incorporated into care.

PLOS Digital Health

**Table 2. Integrating DHTs into Existing Clinical Workflows Presents Significant Informatics Challenges.**

| Subthemes | Exemplar Quotes |
|---|---|
| DHTs are disruptive to clinician workflow | Does primary care take it on? Want to embed everything in primary care? No, we don't have the resources to do it. We don't have the staff. We don't have the space…there's no economic model for that allows us to do this thing. (Participant 4, Division Chief, General Internal Medicine) <br><br> To place the referral electronically, if they're not interfaced with you, it doesn't matter. So, does that take us back into the world of paper faxing and all of those things? So that's one of the barriers, right? And again, it goes back to what technology can we create to seamlessly make that happen, right? Internal to external referrals. So, I think that's something that needs to be explored for the future state (Participant 7 Chief Medical Officer) |
| EHRs are not designed to be interoperable | The EHR companies…they haven't always invested a lot in making this kind of exchange easily accessible, and they put it on the backs of the customers, like, you know, for clinical work. The individual customers … have to pay to get their system to be interoperable. Same with…a lot of these other [EHRs] so that's, that's why we can't exchange information with [another hospital] as an example... because they just never installed that module. There really wasn't a whole lot of interest in developing these kinds of things, especially among the EHR vendors, because it was bad for their business models to be sharing information, because then it makes other vendors more attractive. (Participant 1, Chief Informatics Officer) <br><br> When I try to figure out what, who's prescribing a medication for a patient [the EPIC EHR] usually says unknown provider. It is not made for the ambulatory arena at all. It's terrible, but it's all that we have that communicates with other EHRs. And so, you know, there was all that stuff about interoperability when we [were introduced to] the meaningful use stuff [participant is referring to the Health Information Technology for Economic and Clinical Health (HITECH) Act]. It never really came to pass in a meaningful way. And so instead of having easy to find information so we all know everything about our patients, we have just this waste basket of junk that is really hard to weed through. (Participant 8, Health System President) <br><br> I think that's where you'll need some regulatory purposes, or regulations to come in from a from a governmental standpoint, to say, you know, if you, if you have Medicare beneficiaries, here's a requirement for you within this realm, and you have to enable this interface. (Participant 7, Chief Medical Officer) |
| There are uncharted risks and liabilities associated with digital health information exchange | But I think the other thing is, we're so sensitive to cyber security right now, it is like, you know, everybody's had something bad happen, and pushing data out is pretty easy to keep safe. Bringing something in is where it gets dangerous. Our issues have often been vendors, because our cyber security can be great, but if yours isn't, and you're pushing information into my record, that's where you get in trouble. (Participant 2, Chief Executive Officer) <br><br> A lot of the issues are in the fact that there is not a connection of systems so that the information is not necessarily downloaded into a single system. You have to be connected, and you must give permissions all over the place for the information to be accessible. So clearly within the regulatory environment, I can see where those are definitely issues. (Participant 3, Executive Vice President and Chief Financial Officer) |

## Theme 2: DHTs Present Risks to the Quality of Patient Care

Participants expressed caution about the unintended consequences of DHTs on healthcare delivery, particularly concerning patient safety, equitable access, and continuity of care. This theme included three central subthemes: concerns about patient engagement and trust, information fragmentation, and disparities driven by digital literacy and geographic infrastructure. Table 3 summarizes each subtheme with associated exemplar quotations.

Stakeholders observed that patients may be skeptical of DHTs due to data privacy and security concerns. Without patient trust, adoption and proper use of these tools will be limited, potentially creating gaps between those who can use them and those who cannot. Geographic infrastructure was another source of disparity: in rural areas, limited internet and cellular coverage were cited as major barriers to DHT delivery. Participants also described the digital readiness challenge as a broader downstream consequence of existing structural inequities that are tied to non-clinical drivers of health. Leaders noted that communities with the greatest clinical need for technology-enabled services- including, as an example, rural communities with high prevalences of chronic disease like diabetes and obstructive lung disease- are often those most affected by historical underinvestment in digital infrastructure, unstable housing, lower educational opportunities, and structural racism, all of which impact connectivity and digital device access. These factors were viewed as fundamentally limiting the feasibility of scaling digital tools equitably across the system and getting them in the hands of patients who most need improvements in care access and delivery.

**Table 3. DHTs introduce new risks to the quality of patient care.**

| Subthemes | Exemplar Quotes |
|---|---|
| The impact of digital health is dependent on patient trust | Yeah, I think, I think on the patient side, the issue will be a trust one, right? Because you hear so many cyber events and data being shared, etc., and there's nothing any more personal than healthcare data, right? And so, I think we'll have to you will really, you would really need to partner with the patient advocacy groups to understand and let them be the voice of some of this work, because they won't trust us. They won't trust industry; they won't trust the innovation space unless they have seen it tried and true and that their data is safe. (Participant 8, Health System President) |
| Local infrastructure may create disparities in digital healthcare delivery | We cover all the way from the Mississippi border to the Georgia border, and the Tennessee border down to Birmingham. So, like the top quarter of the state are predominantly where our clients come from. You've got lots of broadband issues and you know cell phone dead zones and that kind of thing as well. (Participant 12, Chief Nursing Officer) |
| Information siloing erodes patient care | I think the biggest risk...is that you wind up diluting your health information across multiple places, and that may lead to lack of coordination of care. The information about what kind of care patients have, particularly if it's preventive care or things that may lower their ultimate risk, is really critical for us to be able to track and monitor. If a patient has diabetes and they have their hemoglobin A1C level done, you know, across town, and we don't have access to that, to that piece of data, then we don't get credit for it, and our numbers look poor. And that ultimately leads to how our performance scores, in terms of the quality ratings of our [Accountable Care Organization] and our value-based programs. (Participant 1, Chief Informatics Officer)<br><br>We want to have as much information about the patients who seek care from [health system] as possible, as humanly possible. So, as a guiding principle, yes, having more data would help and not having comprehensive information hurts us. How would it help? I think inevitably, we want to ensure that we have any information that may be material to a patient's care when they're sick and seek health care. We need to know…everything that's been happening to a patient. (Participant 5, Chief Quality Officer) |

Nearly all participants voiced concerns about the risk of care fragmentation due to the dispersion of health information across disconnected platforms. Participants emphasized that inadequate data sharing can impair providers' ability to deliver integrated, high-quality care and negatively affect performance in value-based payment models. One participant noted:

"The negative is for the PCP or the specialist that the patient engages with…is not to have access to that information if there's an escalation or a worsening in that condition, right? And then if there's worsening, and you don't have access to that data? It undermines…a doctor's function. I don't have the data. I don't know what to do for them unless I get this data, so I don't want to see them until I get more information." (Participant 7, Chief Medical Officer),

Others stressed the critical role of primary care in managing complex patients and warned that displacing this function with condition-specific DHTs could undermine whole-person care and exacerbate the overuse of high-cost services. Participants' accounts of incongruent workflows reflect the CFIR construct of relative priority, as DHTs were perceived as competing with, rather than supporting, established responsibilities and exacerbating inequity. Together, these patterns illustrate how workflow misalignment acts as both a structural and cultural barrier to integration, underscoring the importance of aligning digital tools with existing processes and ensuring adequate organizational support.

### Theme 3: DHTs Expand Access to Care

Participants widely acknowledged the role of DHTs in expanding access to healthcare services, particularly for underserved populations and in resource-limited settings. This theme encompassed five subthemes: the social and cultural advantages of care delivered in the home, improved access across geographic and logistical barriers, the potential to retain patients within a health system by aligning with evolving care preferences, the potential to make care delivery more efficient, and the public health impact of digital services. These subthemes are summarized in Table 4. Several participants emphasized that care delivered in the home has unique emotional and cultural benefits, particularly in communities where family networks provide critical support, underscoring that home-based care can align with community values and

**Table 4. Digital Health Solutions Expand Care Access.**

| Subthemes | Exemplar Quotes |
|---|---|
| Care in the home- social/cultural advantages | And I know this, I'm [of a specific ethnic background] for example, and having the family is there to help is a core value for us. And when your grandmother has an issue, there's no absence of people making sure your grandmother's taking her medicine and then have food she'll eat and all of that stuff. And food insecurity is one of the biggest problems. Well, very few Hispanics go without food when you have the family around, and so they are going to make sure everything is taken care of. So, there's an element of that emotion at home. (Participant 3, Executive Vice President and Chief Financial Officer) |
| Digital services overcome geographical and logistical barriers | Many patients don't have primary care physicians. So, for people who don't have primary care physicians, this creates a care pathway for those people [to access care], and I think that that applies to your Hep C example, or your chronic disease management as well. (Participant 9, Chief Medical Information Officer)<br>Access is a huge issue, especially in a small rural population. We are 1.4 million residents in a rural, spread out, sparsely populated state. Access is a challenge, both from a patient standpoint, driving two hours or more, also from a provider standpoint…We think that leveraging um, uh, digital technology and offering patients access outside of the traditional medical office visit is our best shot at meeting the access needs in the community. We know that there's so much demand. We simply don't have the square footage to bring everyone into the office and to treat them. Geography is not our friend here. (Participant 5, Chief Quality Officer) |
| Patients desire convenient, asynchronous health services | We have a much younger population. We do have a small older adult Medicare population, but a lot of our population…likes to text or email.. (Participant 11, Chief financial officer)<br>We are in competition with some of these programs that are being set up through non physician or non-hospital based or clinic-based entities. And so that's made us more responsive and using telehealth, using extenders. (Participant 8, Health System President) |
| Digital health solutions promote care efficiency | I think in large part it has to serve, serve needs that are not yet already served in a way, in a way that's efficient and effective for patients…Pulmonary rehab is a great one the things that we can't get patients into right now…that they need. That could maybe be done more efficiently if you didn't have to get in your car, drive to a place, do this thing. It has to be able to be safe at home. It has to be efficient at home and has to be effective. But things like, you know, like cardiac rehab, pulmonary rehab, you know, things that could have people sort of weighing in regularly. You know, there's a lot of great tools that, you know, and in my 20-minute office visit…I can't do. (Participant 10, Chief Medical Officer)<br>Primary care access issues, even challenges with specialty access, right, it's a great release valve for a patient to get a test, be able to do it and then get care associated with that. So that's the beauty, right? It's really, really patient centric and consumer friendly. (Participant 6, Chief Medical Officer)<br>So, if your little person has strep throat, right, why can't you, as a mom, do a swab at home and then call into a telemedicine program and get that little person an antibiotic instead of the rigmarole, right? I gotta get my child up. They don't feel well. I gotta take a day out of work, then I gotta get to the doctor's office. So, I think there's so much potential in reducing cost of care through some of these initiatives as well? Because now you've avoided, you know, having to go for a test. I think there's a lot of applicability on what we can do to help decompress the system. (Participant 7, Chief Medical Officer) |
| DHTs may be used to direct public health initiatives | From a public health perspective, I'm 100% on board with the idea of eradicating Hep C. Is that something that is, you know, makes sense for a health system to do, or for all health systems to do? In this kind of way, I think that this type of test to treat program, to me, seems like it's most effective for something finite that doesn't require high-level coordination. (Participant 1, Chief Informatics Officer)<br>And again, it brings me back to that comment of like, public health is a thing. It's a real thing. And if this is a public health initiative, then it should be geared toward reaching more people across the nation, as opposed to, like my 20,000 patients that that were taken care of in [our clinic], for example. And if there are efficiencies to make the public healthier, we should use them in whatever way makes the most sense, because you can't, you can't serve, you know, in the same way, 360 million people in a whole bunch of tiny offices (Participant 4, Division Chief, General Internal Medicine) |

improve adherence and conditions conducive to disease recovery. Participants also consistently described how DHTs enable care delivery for patients who face logistical or geographic barriers to in-person visits. These included individuals without primary care providers, those living in rural areas, and patients unable to attend medical appointments due to personal or professional duties. One participant noted, "Women with children are probably the biggest users of video care, and it's usually off hours." (Participant 7, Chief Medical Officer)

Digital services were also viewed as a means of enhancing patient satisfaction and keeping patients embedded within a single health system. Younger populations in particular were described as preferring asynchronous or virtual communication, such as texting and email. Many participants view telehealth as a competitive necessity, especially in response to emerging non-traditional healthcare entities.

Participants broadly agreed that DHTs can increase system-level efficiency and reduce unnecessary healthcare utilization in brick-and-mortar settings, particularly when deployed for targeted, discrete purposes such as minor illnesses or discrete, standardized chronic care management. Several stakeholders discussed using telehealth or virtual same-day care teams as "release valves" to handle overflow and reduce wait times. DHTs were seen as especially valuable for conditions that require regular monitoring, such as cardiac or pulmonary rehabilitation, areas where traditional office-based models are underutilized or difficult to access. Participants also pointed to the opportunity to reduce healthcare spending by the resources needed for testing and virtual treatment.

Finally, several participants highlighted the value of DHTs in enabling scalable public health interventions. Participants saw home-based digital care as a powerful strategy for addressing finite, high-impact health challenges such as Hepatitis C, where large-scale testing and treatment initiatives could be efficiently coordinated. Others emphasized that DHTs are especially well-suited for broad public health goals rather than individualized care alone. All participants were supportive of national strategies that transcend siloed care models for curable disease screening and treatment and targeted preventative care, such as vaccination administration and documentation.

### Theme 4: Outer Setting Factors Significantly Influence the Implementation of Digital Health Solutions

Participants identified multiple external forces—such as reimbursement policy and market readiness, regulatory constraints, and resources impacting scalability—that critically shape the feasibility and sustainability of digital health initiatives. This theme was organized into three subthemes: reimbursement and payment models, quality control, and scalability challenges. These subthemes are summarized in Table 5.

All stakeholders expressed frustration with the lack of reimbursement pathways for digital health services, particularly in fee-for-service environments. Participants noted that services like virtual pulmonary rehabilitation, which may improve patient outcomes, often reduce hospital utilization or revenue from facility fees and are therefore financially disincentivized, highlighting the conflict between clinical value and financial viability. Others shared that reimbursement is typically limited to face-to-face encounters, limiting innovation even when digital alternatives could improve access and efficiency.

Participants also raised concerns about the consistency and appropriateness of care delivered through third-party digital platforms. They noted that some organizations lacked internal innovation capacity and developing DHTs "in-house," particularly in community settings, isn't feasible or cost-effective, leading them to rely on external vendors. They expressed concerns about the quality of contracted solutions.

One participant stated,

"When you're talking about healthcare tools…one example might be a diabetes nutrition coaching app, things like that, for those types of solutions... we can't home grow this stuff. No way. So, we need to consider industry and how we engage with them safely" (Participant 8, Health System President).

Poor documentation, lack of transparency, and minimal communication with referring providers were cited as potential problems with third-party offerings. Participants also acknowledged that for digital and home-based care programs to be viable, they must operate at a sufficient scale to justify infrastructure investments. Services like hospital-at-home and mobile integrated health were described as financially unsustainable at low patient volumes.

**Table 5. Outer setting factors have a critical impact on the implementation of digital health solutions.**

| Subthemes | Exemplar Quotes |
|---|---|
| Factors external to health systems impact cost-effectiveness and reimbursement | So, you know, how do you provide the best things for your patients when you can't get reimbursed for them? You know, I can't get reimbursed for anything that doesn't involve face to face contact with patients, with the exception of some of our risk contracts that have quality involved...I have facility fees that are visit volume. So, every time a patient shows up, the system gets them, gets money, and any time a professional bills…we get money. (Participant 11, Chief Financial Officer)<br>Virtual pulmonary rehab, which may improve how patient feels, may not be reimbursed like in person rehab. And it may reduce readmission rate, which may have some benefit to the health system. But if it reduces… hopefully this does not come off as callous, but if it reduces acute care visits over the span of that patient, then that may mean less business in a fee for service model. (Participant 3, Executive Vice President and Chief Financial Officer)<br>Medicaid, at least the way the states envision it, it's more in person touch, especially in urban areas. (Participant 6, Medical Group Associate Medical Director)<br>There are codes that support primary care utilizing eConsults, but they're clunky. They're not reimbursed by every payor. The amount of dollars that flow, and I think the highest is $36.00 for an eConsult. So, what we're hearing from the primary care physicians is…I could get my answer pretty quickly, but then the specialist said, hey, you need to order an MRI. You need to do this additional blood work, and that is time that I'm now going to spend on ordering those tests. And then I have to reinterpret them. So that really becomes a demotivator, because there's no reimbursement that's tied to that that supports the added work that I have to do. (Participant 7, Chief Medical Officer) |
| Quality Control | It wasn't the kind of care that we give to our patients….The documentation was bad. The patient engagement was bad…like telling somebody they're newly diagnosed [with a life-changing illness] on January 1st at 1 a.m.… Yeah, so things like that…there were just a myriad of problems with that company. (Participant 12, Chief Nursing Officer)<br>Because, you know, one of the things is, what's the level of quality of care, right, in some of these virtual programs, right? So, I don't know, you know, if I think about the testosterone one [at a men's health clinic]? What kind of workup was done? How appropriate was it? (Participant 7, Chief Medical Officer) |
| Scalability | I think that one of the barriers too is getting, getting to scale for 24/7 365….And if, if you don't have the volume to sustain that, or a revenue source to or hospital, same with hospital at home, then it's really hard to make it work. So, hospital at home lost money. It loses money compared to the bricks and mortar at a census of 12 and under. It's about the same from 12 to 24 it's more profitable than the bricks and mortar at 24 plus. And so that that seems to be one of the challenge of doing these home-based programs. You have to have a mobile workforce. You have to be able to do 24/7 which means you've got a lot of infrastructure investments, and that if you don't use the infrastructure, you can lose a lot of money fast. (Participant 2, Chief Executive Officer) |

## Discussion

Through interviews with key stakeholders across diverse geographic and organizational contexts, we describe both the opportunities and obstacles for implementing DHTs and integrating them with traditional care models. Our findings suggest that while DHTs show promise for expanding access, improving the efficiency of healthcare delivery, and accelerating public health goals, their integration into clinical systems is impeded by a constellation of operational, cultural, financial, and regulatory challenges spanning both the internal and external settings of organizations.

Participants widely acknowledged that DHTs have the potential to transform care delivery, particularly for patients facing barriers to in-person care. Respondents highlighted the benefits of delivering care in the home, aligning with family and cultural values, and reducing geographic and logistical burdens to patients and clinicians. DHTs were described as "release valves" for overburdened health systems, offering timely, patient-centered alternatives for acute needs and chronic disease management. Stakeholders also viewed digital platforms as strategic assets in a competitive healthcare environment, particularly for engaging younger, tech-savvy populations and supporting scalable public health initiatives.

Caution was raised about the propensity of poorly integrated digital programs to further fragment information and sideline primary care. Inadequate documentation and siloed, condition-specific digital interventions could undermine comprehensive, relationship-based care. These issues were especially salient in rural and underserved regions,

where limited digital infrastructure further compounds concerns about equity and widens the digital divide. Participants emphasized broadband access and digital literacy as critical determinants of digital care adoption, reflecting broader structural inequities that shape who can benefit from digital innovation. Consistent with literature on the social and structural determinants of digital health, inadequate connectivity and limited digital skills disproportionately affect communities experiencing longstanding disinvestment and lower socioeconomic opportunity. These upstream factors influence not only patient-level readiness but also system-level decisions related to infrastructure investment, workflow design, and resource allocation [13,15]. Without addressing these structural drivers, as articulated in frameworks such as the Digital Health Equity Framework, digital transformation efforts may unintentionally widen disparities [13,15]. These findings underscore the need for health systems to pair digital innovation with policies that target inequities, including community broadband expansion, access to affordable devices, and integration of digital navigation resources into routine care.

Notably, participants also emphasized that many of the most substantial barriers to digital health adoption exist at the system and policy levels. In existing fee-for-service models, digital interventions that reduce hospital utilization can threaten financial viability, creating structural disincentives to DHTs that directly undermine scalability and long-term sustainability. Reimbursement mechanisms often lag behind innovation, offering little incentive for health systems to invest in the infrastructure, staffing, and workflow redesign required to implement time-saving or quality-improving tools if they are not immediately revenue-generating. When reimbursement pathways are unclear or fragmented, health systems face significant opportunity costs and are less likely to maintain digital services beyond pilot stages, even when clinical value is recognized [30]. This aligns with prior work illustrating how macro-level payment policies shape organizational readiness and technology adoption trajectories [30,31]. Inconsistent regulatory guidance and variable payer support further complicate efforts to scale digital programs by introducing additional financial and compliance uncertainty [31]. Participants also noted that without aligned payment models, promising home-based digital care models must rely on grants or temporary subsidies, making them operationally fragile and financially unsustainable.

Contextualized within the CFIR domains, the primary barriers to implementation of DHTs are within the domains of intervention characteristics (complexity, adaptability, and cost), inner setting (compatibility and available resources), and outer setting (patient needs and resources and external policy and incentives). When aligned with the Expert Complication of Implementation Strategies (ERIC) consensus [32], the recommended implementation strategies with the best evidence include promoting adaptability and compatibility, accessing new funding, and assessing for readiness and identifying barriers and facilitators, the latter of which has been performed as part of this project [32].

These findings align with existing literature highlighting the tension between digital health innovation and system-level readiness [30,31,33,34]. Stakeholders repeatedly pointed to the absence of centralized standards for data interoperability, care quality, and reimbursement as critical barriers to sustainability. Without cohesive policy frameworks, digital health interventions risk remaining fragmented, inconsistently funded, and variably adopted across regions and populations. Federal leadership must consider establishing national infrastructure and regulatory standards that promote secure data exchange, ensure consistent clinical oversight, and align financial incentives with the value delivered by digital care. EHRs in particular may be a target for changes that make them more amenable to interchange with other applications. Much like prior federal efforts to promote electronic health record adoption through Meaningful Use, future legislation could play a pivotal role in creating the technical and financial conditions necessary for digital health to move from pilot programs to durable components of routine care delivery. Health systems alone cannot bear the burden of digital transformation.

Shaped by the COVID-19 pandemic, DHTs have shown promise for advancing public health by enabling scalable, population-level interventions that can be deployed rapidly and efficiently. Home-based diagnostics, apps, and telehealth platforms can support widespread screening, treatment, and follow-up for conditions like treatable infectious diseases. These solutions may expand reach to underserved or geographically dispersed populations, reduce delays in care, and improve data collection for surveillance and response [12,35]. When integrated with public health infrastructure, DHTs can

enhance real-time decision-making, optimize resource allocation, and support more equitable, proactive models of care delivery [10,36]. Such uses would be less revenue and reimbursement-constrained than in discrete health systems and overcome many of their interoperability barriers.

## Limitations

This study has some limitations. Because the Home Test To Treat model was used as the anchoring example, participants may have emphasized features most salient to episodic digital care delivery, such as workflow integration, triage processes, and rapid clinical decision-making. It is possible that using a different digital health technology, such as remote chronic disease monitoring or longitudinal telehealth management, would have surfaced additional considerations. Nonetheless, participants consistently broadened their reflections beyond this model, suggesting that many of the identified organizational challenges reflect system-wide readiness rather than characteristics of a single digital tool.

The sample consisted of senior leaders from U.S.-based health systems, which may limit the generalizability of the analysis. Frontline clinicians, patients, and representatives from international settings were not included as participants. Participants' views may be influenced by their institutional role, access to resources, or previous experiences with digital health implementation. Additionally, there is a possibility of sampling bias due to non-response from one potential participant. Nonetheless, the diversity of roles and regions represented enhances the credibility of the findings, and the use of rigorous qualitative methods supports the reliability of the themes identified.

## Conclusion

Digital health solutions hold significant promise to improve care access, efficiency, and patient engagement. However, without targeted investment in infrastructure, interoperability, regulatory, and reimbursement reform, these tools risk exacerbating care fragmentation and inequity as well as non-adoption by clinicians and patients. Sustainable adoption will require deliberate alignment of clinical, operational, and policy priorities, with continuous input from the patients and clinician teams they are meant to serve.

## Supporting information

**S1 Appendix. Interview Guide.**
(DOCX)

## Author contributions

**Conceptualization:** Laurel O'Connor, Andrew C. Weitz, Apurv Soni.

**Data curation:** Apurv Soni.

**Formal analysis:** Laurel O'Connor, Leah Dunkel, Peter K. Lindenauer, Apurv Soni.

**Funding acquisition:** Apurv Soni.

**Investigation:** Laurel O'Connor, Apurv Soni.

**Methodology:** Laurel O'Connor, Leah Dunkel, Andrew C. Weitz.

**Project administration:** Laurel O'Connor, Leah Dunkel.

**Supervision:** Laurel O'Connor, Allan Walkey.

**Writing – original draft:** Laurel O'Connor.

**Writing – review & editing:** Leah Dunkel, Andrew C. Weitz, Allan Walkey, Peter K. Lindenauer, Apurv Soni.

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
