## [Decision Letter · Decision Letter 0]

24 Nov 2025

Response to Reviewers
Revised Manuscript with Track Changes
Manuscript
**Journal Requirements:**

1. Please provide an Author Summary. This should appear in your manuscript between the Abstract (if applicable) and the Introduction, and should be 150–200 words long. The aim should be to make your findings accessible to a wide audience that includes both scientists and non-scientists. Sample summaries can be found on our website under Submission Guidelines: [LINK]

https://journals.plos.org/digitalhealth/s/submission-guidelines#loc-parts-of-a-submission

**Additional Editor Comments (if provided):**
**Reviewers' Comments:**

**Comments to the Author**

1. Does this manuscript meet PLOS Digital Health’s publication criteria?

Reviewer #1: Yes

Reviewer #2: Yes

2. Has the statistical analysis been performed appropriately and rigorously?

Reviewer #1: Yes

Reviewer #2: Yes

3. Have the authors made all data underlying the findings in their manuscript fully available (please refer to the Data Availability Statement at the start of the manuscript PDF file)?

Reviewer #1: Yes

Reviewer #2: Yes

4. Is the manuscript presented in an intelligible fashion and written in standard English?

Reviewer #1: Yes

Reviewer #2: Yes

Reviewer #1: Overall Evaluation

This qualitative study provides valuable insights into the facilitators and barriers of implementing digital health technologies (DHTs) within diverse U.S. health systems. The manuscript is methodologically sound and clearly structured, employing the CFIR and CENTER-IT frameworks to guide data collection and analysis. The inclusion of senior system-level stakeholders enhances the policy relevance and practical significance of the findings.

However, several issues regarding conceptual clarity, methodological transparency, and stylistic consistency should be addressed before publication. Once revised, the paper will make a meaningful contribution to the literature on digital health implementation readiness.

Major Comments

1. Clarify the conceptual relationship between CENTER-IT and CFIR (Methods, Lines 96–109).

The description of how CENTER-IT relates to CFIR is somewhat dense. Readers unfamiliar with CENTER-IT may not understand its unique contribution.

Recommendation: Add a concise explanation or visual summary differentiating CENTER-IT as a participatory adaptation methodology grounded in CFIR, emphasizing its multi-level stakeholder engagement.

2. Improve sampling transparency and reflexivity (Methods, Lines 117–123, 160–166).

Details on how participants were identified and diversity ensured are limited. Reflexivity regarding the physician interviewers should also be acknowledged.

Recommendation: Clarify recruitment rationale and diversity (academic vs. community systems) and include a brief reflexivity statement describing how investigators’ professional backgrounds may have influenced data interpretation.

3. Deepen analytical interpretation in Results (Themes 1–2)

Quotations are rich but occasionally overshadow analysis. The link between data and CFIR constructs could be more explicit.

Recommendation: Strengthen interpretive commentary connecting participant quotes to CFIR constructs (e.g., complexity, compatibility) and ERIC strategies to highlight practical implications.

4. Expand discussion on policy and economic implications (Discussion, Lines 347–372).

Reimbursement and regulatory barriers are described but not fully analyzed.

Recommendation: Link financial disincentives to implementation outcomes (e.g., scalability, sustainability) and situate findings within health-economic and policy implementation literature (e.g., Cresswell 2013; Greenhalgh 2017).

5. Ensure terminology and stylistic consistency

Terms such as digital health tools, digital solutions, and digital health technologies are used interchangeably.

Recommendation: Adopt a single term—preferably Digital Health Technologies (DHTs)—throughout the manuscript.

Minor Comments

1. Limitations: Please note that frontline clinicians and patients were not included in the study sample.

2. Methods – Setting:

Original: “The study staff was based at an urban academic tertiary care medical center in Massachusetts that serves urban, suburban, and rural communities.”

Issue: The verb “was” should be changed to “were” because the subject staff is plural.

3. Methods – Participant Recruitment:

Original: “expanded this list during recruitment..8, 17, 20”

Issue: There is a typographical error (double period “..8”), and the placement of the reference numbers is incorrect. Please correct the punctuation and reference positioning.

4. References:

o Remove the residual citation marker {Moser, 2018 #74}.

o Correct capitalization: “krippendorff K.” → “Krippendorff K.”

o Correct the journal title: “he Qualitative Report” → “The Qualitative Report.”

Overall Recommendation

Recommendation: Major Revision

This manuscript addresses a timely and important topic with strong methodological potential. With clearer theoretical positioning, enhanced reflexivity, and consistent terminology, the paper will make a valuable contribution to understanding system-level readiness for digital health implementation.

Reviewer #2: This manuscript presents a timely and well-conducted qualitative study exploring health system leaders’ perspectives on the implementation of digital health technologies (DHTs) across diverse U.S. settings. The research addresses a critical gap in implementation science by focusing on systemic readiness—rather than just technological feasibility—and offers nuanced insights into the interplay between clinical workflows, equity considerations, policy constraints, and financial sustainability. The use of the Consolidated Framework for Implementation Research (CFIR) and the CENTER-IT methodology strengthens the rigor and relevance of the findings. The paper is clearly written, thematically rich, and contributes meaningfully to ongoing discourse about scaling digital health equitably and effectively.

Suggestions for Revision

1. Clarify the Role of the “Home Test to Treat” Example

The manuscript introduces the Home Test to Treat program as a concrete anchor for interviews. However, its role in shaping responses could be more explicitly discussed:

Consider adding a brief methodological note explaining whether participants’ views were primarily shaped by this example or generalized beyond it.

In the Discussion, briefly reflect on whether findings might differ if a different DHT (e.g., remote monitoring for chronic disease) had been used as the prompt.

2. Expand on Equity Considerations

While disparities related to broadband access and digital literacy are noted, the analysis could deepen its engagement with structural inequities:

Explicitly connect “techquity” concerns to social determinants of health (e.g., income, education, race/ethnicity).

Consider citing recent frameworks like the Digital Health Equity Framework (Richardson et al., npj Digit Med 2022) to strengthen conceptual grounding.

3. Refine Terminology Consistency

The term “digital health technologies (DHTs)” is well-defined, but at times the manuscript uses “digital tools,” “digital solutions,” and “decentralized care” interchangeably. Ensure consistent use of core terminology, especially in the Results and Discussion.

Clarify early on whether “integration” refers specifically to EHR interoperability or broader workflow/system alignment.

4. Address Limitations More Fully

The current limitations section appropriately notes the focus on senior leaders. Consider adding:

Potential bias from non-response (1 of 13 approached declined); while minimal, it’s worth acknowledging.

The absence of patient or frontline clinician voices limits insight into end-user experience—a point that could be tied to future research directions.

5. Minor Editorial Suggestions

In Table 2, the quote from Participant 8 appears cut off mid-sentence (“President)”). Please verify transcription accuracy.

Ensure all acronyms (e.g., ACO, ERIC) are defined at first use.

The phrase “Implementation Science: Health System Science” in Keywords uses a colon; consider using a semicolon or listing as separate terms per journal style.

**Do you want your identity to be public for this peer review?** For information about this choice, including consent withdrawal, please see our Privacy Policy

Reviewer #1: No

Reviewer #2: No

**Figure resubmission:**

**Reproducibility:** To enhance the reproducibility of your results, we recommend that authors of applicable studies deposit laboratory protocols in protocols.io, where a protocol can be assigned its own identifier (DOI) such that it can be cited independently in the future. Additionally, PLOS ONE offers an option to publish peer-reviewed clinical study protocols. Read more information on sharing protocols at https://plos.org/protocols?utm_medium=editorial-email&utm_source=authorletters&utm_campaign=protocols

---

## [Decision Letter · Decision Letter 1]

24 Dec 2025

Digitally Delivered, Systemically Challenged: A Qualitative Study of Health System Readiness for Digital Care

PDIG-D-25-00455R1

Dear Dr. O'Connor,

We are pleased to inform you that your manuscript 'Digitally Delivered, Systemically Challenged: A Qualitative Study of Health System Readiness for Digital Care' has been provisionally accepted for publication in PLOS Digital Health.

Best regards,

Hadi Ghasemi

Academic Editor

PLOS Digital Health

**Additional Editor Comments (if provided):**

**Reviewer Comments (if any, and for reference):**

Reviewer's Responses to Questions

**Comments to the Author**

Reviewer #1: All comments have been addressed

Reviewer #2: All comments have been addressed

publication criteria?

Reviewer #1: Yes

Reviewer #2: Yes

3. Has the statistical analysis been performed appropriately and rigorously?

Reviewer #1: Yes

Reviewer #2: Yes

4. Have the authors made all data underlying the findings in their manuscript fully available (please refer to the Data Availability Statement at the start of the manuscript PDF file)?

Reviewer #1: Yes

Reviewer #2: Yes

5. Is the manuscript presented in an intelligible fashion and written in standard English?

Reviewer #1: Yes

Reviewer #2: Yes

Reviewer #1: All comments have been adequately addressed.

Reviewer #2: The manuscript, in its current revised form, is technically sound, ethically rigorous, and makes a meaningful contribution to the fields of digital health and implementation science.

The authors have been exceptionally responsive to the reviewers' feedback. The revisions have substantially strengthened the manuscript, as evidenced by� enhanced conceptual clarity�deeper analysis, terminology consistency and comprehensive addressing of limitations.

**Do you want your identity to be public for this peer review?** For information about this choice, including consent withdrawal, please see our Privacy Policy

Reviewer #1: **Yes: ** Shizhu Bai

Reviewer #2: No
